# Skin-like cryogel electronics from suppressed-freezing tuned polymer amorphization

Xiansheng Zhang[1,10], Hongwei Yan[2,10], Chongzhi Xu[3,10], Xia Dong [4], Yu Wang[4], Aiping Fu[1], Hao Li[5], Jin Yong Lee [5], Sheng Zhang [6], Jiahua Ni[7], Min Gao[8], Jing Wang [8], Jinpeng Yu[2], Shuzhi Sam Ge [9], Ming Liang Jin [2]✉, Lili Wang [1]✉ & Yanzhi Xia[1]

The sole situation of semi-crystalline structure induced single performance remarkably limits the green cryogels in the application of soft devices due to uncontrolled freezing field. Here, a facile strategy for achieving multi-functionality of cryogels is proposed using total amorphization of polymer. Through precisely lowering the freezing point of precursor solutions with an antifreezing salt, the suppressed growth of ice is achieved, creating an unusually weak and homogenous aggregation of polymer chains upon freezing, thereby realizing the tunable amorphization of polymer and the coexistence of free and hydrogen bonding hydroxyl groups. Such multi-scale microstructures trigger the integrated properties of tissue-like ultrasoftness (Young's modulus <10 kPa) yet stretchability, high transparency (~92%), self-adhesion, and instantaneous self-healing (<0.3 s) for cryogels, along with superior ionic-conductivity, antifreezing (−58 °C) and water-retention abilities, pushing the development of skin-like cryogel electronics. These concepts open an attractive branch for cryogels that adopt regulated crystallization behavior for on-demand functionalities.

Although hydrogels for nascent applications, such as ionic skin[1,2], wearable sensors[3–5], and optical devices[6,7], primarily rely on polymers with at least one chemical network due to their inherent flexibility and integrated functions, their chemical residuals (such as unreacted monomers or initiators) in the final materials are unavoidable, causing serious environmental or health concerns. Thus, the entire physical hydrogels, particularly the well-known cryogels from the freeze–thawed (FT) method, have become ideal candidates for the aforementioned applications, and have garnered increasing attention owing to their nontoxicity, simple fabrication without harsh synthetic condition, biocompatibility, and their similarity to biological tissues[8,9]. However, green cryogels with single-polymer system suffer from extremely restricted functions. For these cryogels, sharing the combinational properties (such as flexibility, transparency, self-adhesiveness, and self-healing) is almost impossible but remains the prerequisite for emerging applications[10,11].

[1]State Key Laboratory of Bio-Fibers and Eco-Textiles, College of Textiles and Clothing, Qingdao University, 266071 Qingdao, China. [2]Institute for Future, Shandong Key Laboratory of Industrial Control Technology, School of Automation, Qingdao University, 266071 Qingdao, China. [3]College of Materials Science and Engineering, Qingdao University, Qingdao 266071, China. [4]CAS Key Laboratory of Engineering Plastics, Institute of Chemistry, Chinese Academy of Sciences, 100190 Beijing, China. [5]Department of Chemistry, SungKyunKwan University, Suwon 16419, Korea. [6]State Key Laboratory of Fluid Power and Mechatronic Systems, College of Mechanical Engineering, Zhejiang University, 310027 Hangzhou, China. [7]College of Biological Science and Medical Engineering, Donghua University, 201620 Shanghai, China. [8]Institute of Environmental Engineering, ETH Zürich, Zürich 8093, Switzerland. [9]Department of Electrical and Computer Engineering, National University of Singapore, 117576 Singapore, Singapore. [10]These authors contributed equally: Xiansheng Zhang, Hongwei Yan, Chongzhi Xu. ✉e-mail: jinmingliang@qdu.edu.cn; llwang@qdu.edu.cn

More recently, tremendous efforts have been devoted to creating well-organized hierarchical micro/nanostructures in inherent polymer networks to enhance the mechanical strength and toughness of cryogels for tissue engineering applications. For example, inducing the aggregation of polymer chains through salting-out effect[12], producing high crystallinity through freeze-drying and annealing treatments[13], forming an anisotropic structure through mechanical training[14], or directional freezing method[15]. Despite these exciting advances, some important issues remain for cryogels in their emerging applications. First, the existing mechanism of enhancing the heterogeneous aggregation of polymer chains or the degree of crystallinity pushes the mechanical strength to a high value but with a narrow range and further sacrifices the unique advantages of hydrogels, including flexibility, extendibility, transparency, and high water contents, which severely limits their application in new areas. For instance, the modulus range of soft tissue is about 1–1000 kPa[16], wherein the high modulus region (>100 kPa), corresponding to those in robust tissues (tendon or cartilage), can be satisfied by the aforementioned stiff cryogels. Notably, expanding the modulus of green cryogels into an ultralow modulus region (i.e., 1–100 kPa, particularly for the ultra-softness of 1–10 kPa) has rarely been of focus and it is highly desirable for soft-tissue-matching materials[17–19] (such as artificial skin and flexible wearable devices). Second, characterized by rich supramolecular sites (such as typical polyvinyl alcohol, gelatin, and alginate), cryogels are expected to exhibit some distinctive features such as self-adhesiveness and self-healing. However, existing cryogels are nonstick and only partially self-healed under high polymer concentration[20]. This is because the construction of a physical network consumes most of the free supramolecular sites and reduces the mobility of polymer chains. Third, although creating anisotropic hydrogels with well-aligned polymer chains through mechanical stretching[14] has become popular in mimicking skin-like J-shaped stress–strain behaviors, the adjustment of polymer conformation for cryogels anchored by crystalline points is difficult. Their poor extendibility considerably restricts the applied strain upon stretching, substantially increasing the difficulty in producing cryogel-based tissue-like materials. Based on these aspects, the sole semi-crystalline cross-linked network produced from the FT process becomes the bottleneck in achieving all appealing properties, such as flexibility, transparency, self-adhesiveness, self-healing, and high anisotropy for cryogels. This necessitates an urgent need for developing new multi-scale microstructures via a distinctive forming mechanism in physical cryogels that can trigger their structural merits to a maximum extent. This process should be promising for achieving multifunctionality and satisfy the stringent requirements of emerging applications. However, it is considerably challenging, because the easily obtained semi-crystalline structures of cryogels, to our knowledge, have rarely been altered via the green FT method since their discovery in the 1970s by Peppas[9].

In cryotropic gelation[9,21], freezing-induced ice crystals result in phase separation of the polymer, wherein the polymer crystallization with the formation of supramolecular bonds occurs in the polymer-rich region and these physical interactions remain intact after thawing, serving as physical cross-linking points. Herein, we attempt to make a breakthrough in the microstructure of cryogels by regulating the crystallization behavior of water molecules from the source. To achieve this, rather than enhancing the polymer aggregation as usual, a distinctive route is proposed herein. If the ice crystal is suppressed at the initial freezing stage of the gelation, its expelling effect on polymer chains and the resultant liquid–liquid phase separation (polymer-rich and polymer-poor phases) will be greatly weakened, which is promising for producing a relatively weak yet homogenous aggregation of polymer chains. Thus, the amorphization of the polymer network may be achieved along with the simultaneous release of a part of free supramolecular sites (such as hydroxyl groups) because of the large distance between adjacent chains. Based on these particular

hierarchical structures at multi-length scales, the potential multifunctionality of cryogels may be triggered according to the structure–property theory, which is the interest of our work. In this assumption, the well-controlled suppressing crystallization of water below freezing is crucial; however, it is a major obstacle because of the difficulty in precisely controlling the external cryogenic field and the rapid growth of ice.

In this work, instead of changing the cryogenic temperature ($T_c$), we successfully tackle this issue by lowering the freezing temperature ($T_f$) of water molecules, assisted by an anti-freezing salt ($CaCl_2$)[22]. Through regulating the temperature gap ($\Delta T = T_c - T_f$) to various extents, the precise control of ice size in a wide range is successfully achieved. To test our hypothesis, using typical polyvinyl alcohol (PVA) cryogel as a model system, a facile suppressed-freezing thawing (SFT) method is innovatively proposed. Specifically, upon freeze–thaw process, the growth of ice crystals is remarkably suppressed, subsequently inducing a homogenous aggregation of polymer chains, notably achieving the tunable amorphization of the cross-linked network and the coexistence of free and H-bonded hydroxyl groups. These hierarchical microstructures endow the simple recipe hydrogels (the new class of cryogel is named as suppressed cryogel) with the integrated properties of tissue-like ultrasoftness (Young's modulus of 4–10 kPa) along with stretchability (~600%), transparency (~92%), self-adhesiveness, instantaneous self-healing (<0.3 s), and remoldability. Furthermore, they exhibit superior ionic conductivity (12.94 S m$^{-1}$, 20 °C) as well as antifreezing (–58 °C) and water-retention ability with the aid of $CaCl_2$. Notably, these attributes can be adjusted within a wide range by varying the degree of amorphization. Thus, with these appealing traits together, we demonstrate the capacity of using single suppressed cryogels in a variety of advanced applications, such as artificial nerve fibers, touch panels, and pressure sensors, as confirmed by multiple proof-of-concept demonstrations. Thus, the emergency of suppressed-freezing-tuned amorphization of cryogels in this research aids in developing a new generation of skin-like cryogel electronics.

## Results

### Design principle of the suppressed cryogels

Following the SFT strategy, an anti-freezing agent $CaCl_2$ was selected as the ice size regulator to realize the tunable suppression of ice growth upon freezing, based on the following reasons. First, the incorporation of $CaCl_2$ into water transforms some free water molecules into bound water molecules, inhibiting the original hydrogen bonding network between water molecules and the formation of ice crystals during cooling (Supplementary Fig. 1a). Following this anti-freezing mechanism, the introduction of $CaCl_2$ into the precursor solution of hydrogels could reduce its freezing temperature ($T_f$), thereby widely altering the extent of depression (0 to –58 °C) by varying the concentration of $CaCl_2$ (Fig. 1a). Second, $CaCl_2$ exhibits salting-in effects in polymer chains[16], guaranteeing the solubility of polymer in water, particularly in case of high-concentration PVA solutions. Third, compared with other antifreezing agents of organic solvents (e.g., ethylene glycol[23,24], dimethyl sulfoxide[25,26]), the suppressed-freezing hydrogels in this research remain environmental/health friendly even with high $CaCl_2$ concentration. Based on these facts, the phase diagram (Fig. 1a) of the aqueous PVA/$CaCl_2$ solution (the concentration of PVA, $C_{PVA} = 14$ wt%) is established, depending on the freezing point of precursor solution collected using the differential scanning calorimetry and their freezing states at different cryogenic temperatures ($T_c$) (Supplementary Fig. 1b, c). Here, three possible principles for designing cryogels are obtained, depending on the cryogenic states at the widely used $T_c$ of −20 °C. (1) With low $CaCl_2$ concentrations (e.g., $C_{CaCl2} = 0$, 1 M; the blue region in Fig. 1a), the $T_c < T_f$ results in large ice crystals during the cryogenic process, indicating the formation mechanism of conventional cryogels (Fig. 1b–d). (2) With sufficient $CaCl_2$ (e.g., $C_{CaCl2} = 2$, 3, 3.5 M; red region in Fig. 1a), a

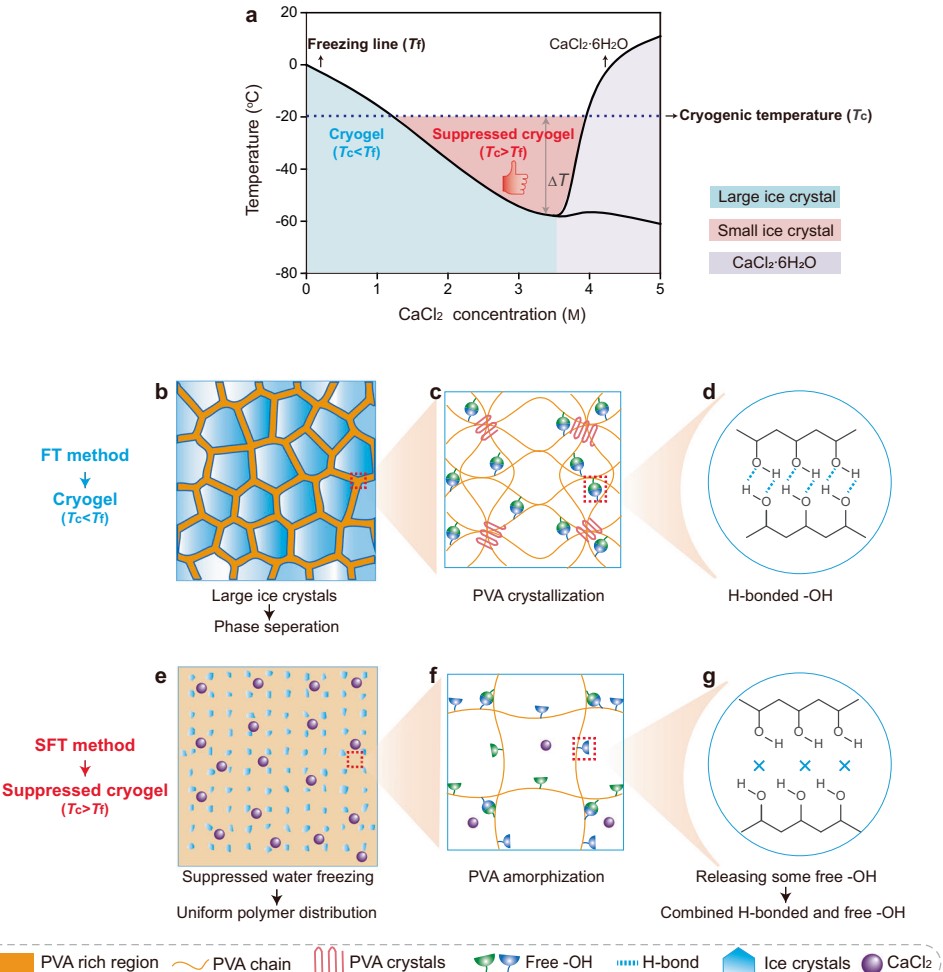

**Fig. 1 | Schematics of the design principle for suppressed cryogels. a** Phase diagram of the PVA/CaCl$_2$/H$_2$O solution, and the design principle of conventional cryogels from freeze-thawed (FT) method and suppressed cryogels from suppressed-freeze-thawed (SFT) method based on the relationship between freezing temperature $T_f$ and cryogenic temperature $T_c$ ($\Delta T = T_c - T_f$). **b–g** The cryogenic state and the corresponding multi-scale structures for conventional cryogels (**b–d**) and suppressed cryogels (**e–g**).

solution freezing temperature lower than cryogenic temperature ($T_c > T_f$) becomes available, successfully weakening the crystallization of water. At this state, within a critical $\Delta T$ ($T_c - T_f$), a small amount of free water molecules of precursor solution still rearrange into the ice but with an ultrasmall size upon freezing, as indicated by its transparent but nonflowing state (Supplementary Fig. 1d), successfully creating a new generation of suppressed cryogels (Fig. 1e–g). Notably, the extent of suppression for ice could be regulated by changing $\Delta T$, as determined by $T_f$ (related to $C_{CaCl2}$) and $T_c$. The suppressed cryogels with this formation principle are the focus of this work. (3) With considerably high CaCl$_2$ concentrations (e.g., $C_{CaCl2}$ = 4, 5 M; the purple region in Fig. 1a), the precursor solution becomes opaque upon freezing (Supplementary Fig. 1c) because of the formation of CaCl$_2$·6H$_2$O[22]; however, the resultant hydrogels after thawing are unstable and dissolved at ambient temperature. This instability may be due to the considerable suppression of water crystallization under large $\Delta T$, causing a loosely cross-linked network and a large distance between the adjacent PVA chains.

## Formation of amorphous multi-scale structures

Using the SFT strategy, a series of distinctive microstructures is constructed at multi-length scales in the suppressed cryogels (from large scale to small scale in Fig. 2). First, in combination with small-angle and wide-angle X-ray scattering (SAXS/WAXS), the growth of crystalline domain for hydrogels, serving as crosslinking points, was revealed, determining their microscopic properties. With an increase in CaCl$_2$ concentrations, the peak intensity in both curves becomes weak. In particular, the diffraction peak at $2\theta = 15.5°$ ($d = 4.63$ Å) in WAXS patterns (8°–18°), corresponding to the (10$\bar{1}$) reflection of crystalline PVA[27], is obvious in cryogels ($C_{CaCl2}$ = 0, 1 M); however, it completely disappears for the suppressed cryogels ($C_{CaCl2}$ = 2, 3 M; Fig. 2c), confirming the successful amorphization of semi-crystalline cryogels using the SFT strategy. According to the SAXS patterns (Fig. 2a), with the increase of CaCl$_2$ concentration, the diffraction circle gradually moves toward a beamstop and the peak value $q_{max}$ of the $Iq^2$–$q$ curves shifts to a lower $q$ region. These variations imply that a larger long period (derived from Bragg's law $L = 2\pi/q_{max}$, reflecting the average distance between neighboring crystallites) is favored in the suppressed cryogels (Fig. 2b). For example, the average crystallite spacing of cryogels $L_{0M}$ is estimated to be 15.1 nm, consistent with the reported values[28,29], and becomes double sized with 1 M CaCl$_2$ ($L_{1M}$ = 31.4 nm). Notably, with CaCl$_2$ concentrations of >2 M (suppressed cryogels), the diffraction circles of SAXS patterns are buried within a beamstop (Supplementary Fig. 2), and thus the $L$ of these samples are roughly depicted based on the variation trend (dotted box in Fig. 2b), instead of calculating with the Bragg's law. The pronounced increase of $L$ for suppressed cryogels is ascribed to the substantially reduced crystallinity, where considerably more interstitial amorphous chains are

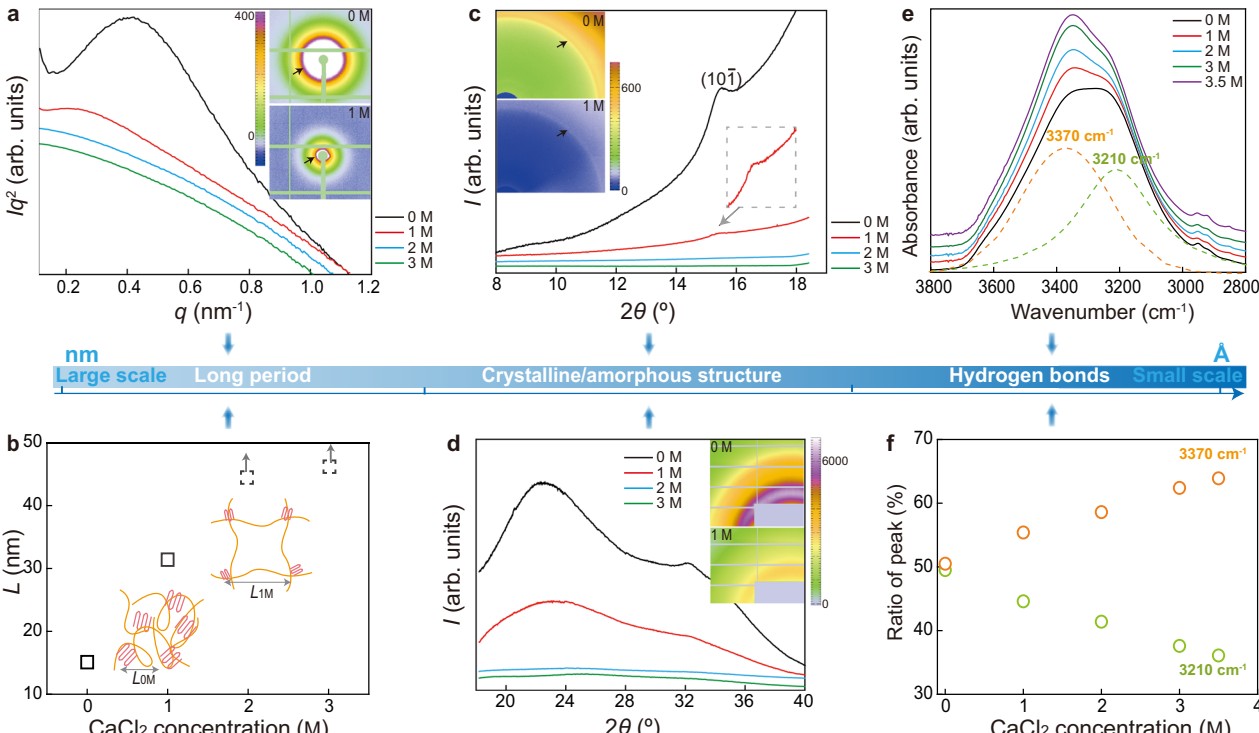

**Fig. 2 | Microstructural characterizations of cryogels and suppressed cryogels at multi-length scales. a** The $Iq^2-q$ curves and SAXS patterns ($I$ and $q$ represent the scattering intensity and scattering vector, respectively). **b** Variations of long-period $L$ with CaCl₂ concentrations. **c, d** WAXS patterns and their integrated curves in the range of 8°–18° (**c**) and 18°–40° (**d**). **e, f** ATR-FTIR spectra (**e**) and the ratio of each peak centered at 3370 and 3210 cm⁻¹ to the sum of both peaks by the peak fitting method (**f**).

loosely distributed, thereby enlarging the distance between adjacent periodic units. Thus, based on SAXS and WAXS (8°–18°) analysis, it could be concluded that assisted by antifreezing CaCl₂, suppressed ice crystals upon freezing make the PVA chains undergo a limited aggregation from each other, avoiding the aggregation-induced crystallization of polymers and creating an entirely amorphous yet relatively homogenous network in suppressed cryogels. In addition, the peaks centered at $2\theta = 22.4°$ and 32.2° in the WAXS patterns (18°–40°) are assigned to the diffraction of free water, and the corresponding integrated areas can be used to evaluate the amount of free water in the hydrogels[27]. Notably, as the CaCl₂ concentration increases to 2 M, these peaks rapidly disappear, demonstrating that most water molecules in the suppressed cryogels stay bounded with salt, guaranteeing the suppressed crystallization of water molecules upon freezing.

Furthermore, the hydrogen bonding interaction of PVA hydrogels was characterized using Fourier transform infrared spectroscopy (FTIR). The vibration band around 3300 cm⁻¹ is separated into two bands of 3370 and 3210 cm⁻¹ via the peak fitting method (Fig. 2e), which are assigned to the stretching vibration of free –OH groups and H-bonding –OH groups between the PVA-PVA chains, respectively[30]. According to the ratio of each peak (Fig. 2f), with an increase in CaCl₂ concentration, the peak ratio at 3210 cm⁻¹ is almost linearly reduced, whereas the peak ratio at 3370 cm⁻¹ is increased. These variations convince us that in comparison with cryogels, the H-bonding interaction of the suppressed cryogels becomes weak, transforming part of the H-bonding –OH groups into free –OH groups. This is reasonable because the SFT strategy induces a limited aggregation of PVA chains during freezing and a large distance between the adjacent chains makes free –OH groups available. Furthermore, the molecular dynamic simulation indicates that, compared with conventional cryogels, the suppressed cryogels favor a more extended conformation of polymer chains, and the corresponding H-bonding interaction between polymer chains is reduced (Supplementary Fig. 3). Thus, it is evident that

the SFT strategy creates uniquely hierarchical microstructures of a homogenously amorphous network with combinational H-bonding and free –OH groups, thereby breaking the structural limit of semi-crystalline cryogels with H-bonding dominated –OH groups.

Notably, the arrangement of the polymer chains could be widely adjusted via altering the CaCl₂ concentration, which precisely controls $\Delta T$ (Fig. 1a) or the antifreezing ability of water.

**Integrated properties of suppressed cryogels**

The amorphous hierarchical structures endow PVA hydrogels with a series of exciting performances. The first impressive property of the suppressed cryogels obtained using the SFT strategy is their high transparency (Fig. 3a and Supplementary Fig. 4), wherein a transmittance of 91.6% is achieved at a wavelength of 600 nm, exhibiting a sharp contrast with the completely opaque cryogels (0.4%). The transparency feature is contributed by the homogenous yet amorphous structure of the suppressed cryogels. More importantly, after soaking in liquid nitrogen (−196 °C), the suppressed cryogels maintained a transparent and intact appearance, whereas the cryogels became more white and immediately fractured. When they were placed back into ambient temperature (24 °C), the suppressed cryogels were converted to a flexible state within a considerably short time of 40 s, compared with the long period of 420 s for cryogels (Fig. 3b and Supplementary Movie 1). The freezing-tolerant transparency arises from that with sufficient ions bound with water molecules, a very small number of free water transforms into ultra-small-sized ice crystals. The greatly reduced mobility of polymer chains in liquid nitrogen makes the hydrogel rigid, and it can be rapidly converted back once the hydrogels are put back in the ambient condition. These characters demonstrate the potential applications of suppressed cryogels as optical materials, particularly in ultracold environments. For example, the texture of leaves is visible when covered with a suppressed cryogel during the cold winter in northern China (Supplementary Fig. 4b).

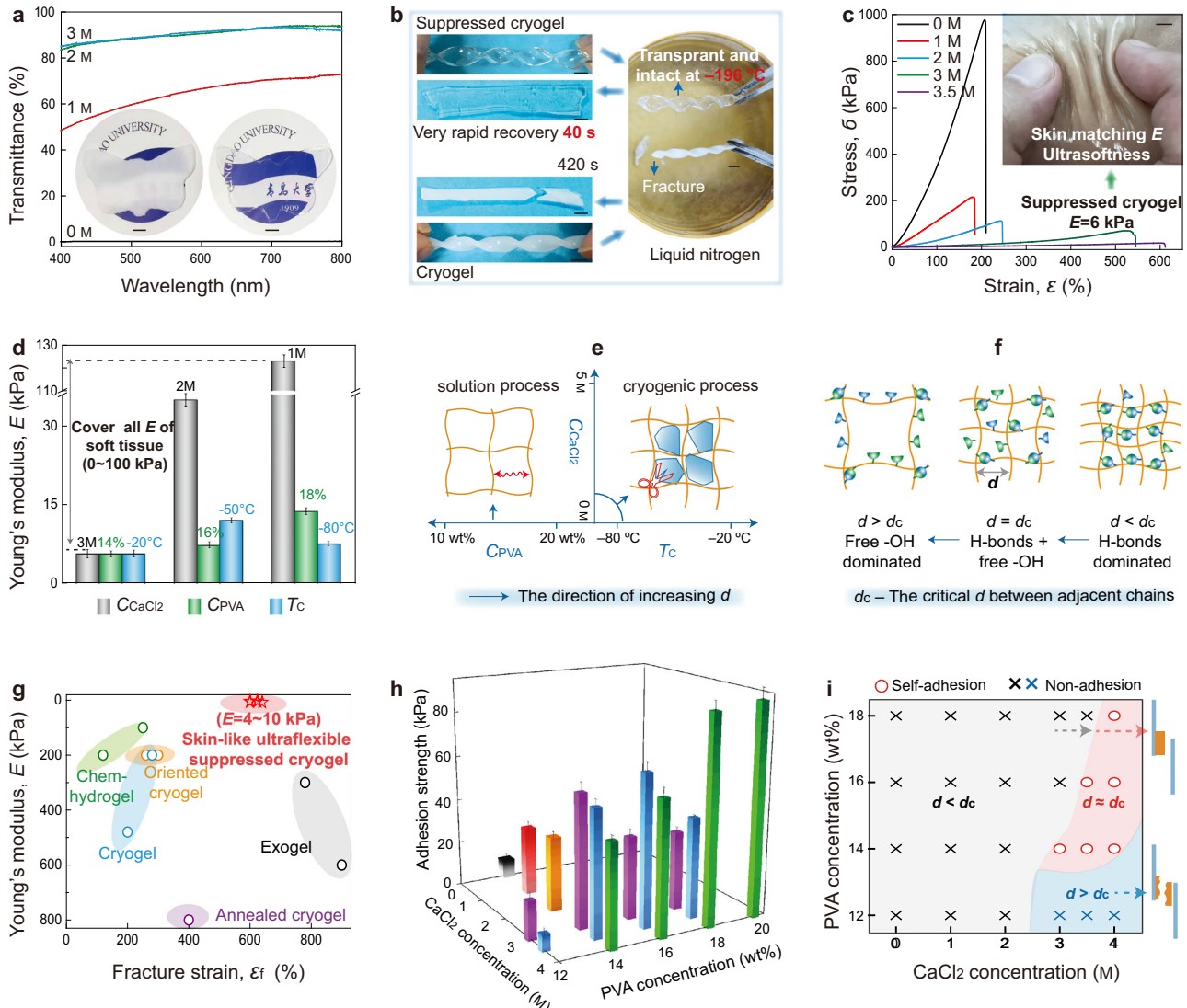

**Fig. 3 | Integrated properties of the suppressed cryogels. a** Transmittance of cryogels and suppressed cryogels with a thickness of 2 mm (inset: the photographs of opaque cryogels and transparent suppressed cryogels (3 M)). **b** The twisted cryogels and suppressed cryogels are placed into liquid nitrogen (−196 °C) and then converted back into ambient temperature (24 °C). **c** Stress–strain curves and the image of flexible suppressed cryogels (ultralow Young's modulus $E$) closely contacted with skin. **d** Variations in Young's modulus with $CaCl_2$ concentration ($C_{CaCl2}$) ($C_{PVA}$ = 14 wt%, $T_c$ = −20 °C), PVA concentration ($C_{PVA}$) ($C_{CaCl2}$ = 3 M, $T_c$ = −20 °C) and cryogenic temperature ($T_c$) ($C_{PVA}$ = 14 wt%, $C_{CaCl2}$ = 3 M). **e, f** Schematic showing the effect of $C_{PVA}$, $C_{CaCl2}$, and $T_c$ on the microstructure of cryogels upon FT process (**e**), including the variation of free/H-bonding −OH groups with the distance between adjacent chains $d$ (**f**). **g** Comparison of Young's modulus versus fracture strain among different PVA hydrogels, including those of the chemically cross-linked hydrogel (chem-hydrogel), the cryogel treated using directional ice-template or mechanical training (oriented cryogel), the cryogel treated using annealing (annealed cryogel), the hydrogel fabricated using solvent exchange (exogel), and the cryogel created using the FT and SFT strategies (i.e., the cryogel and suppressed cryogel in this work). **h, i** Variations of adhesion strength with $C_{PVA}$ and $C_{CaCl2}$ under $T_c$ = −20 °C (the adhesion strength of <50 kPa was defined as non-adhesive nature.) Error bars = standard deviation ($n$ = 6) in (**d, h**). Scale bars: 5 mm in (**a–c**).

Except for the high transparency, the suppressed cryogels display a distinct mechanical behavior with an ultrasoft yet stretchable nature. According to the stress–strain curves (Fig. 3c), with increasing $CaCl_2$ concentration, Young's modulus ($E$) of the obtained hydrogels is considerably reduced, while the fracture strain ($\varepsilon_f$) considerably increases. Among them, the typically suppressed cryogels with 3 M $CaCl_2$ exhibit an $E$ of 5.6 kPa, which is almost two orders of magnitude decrease in comparison to that of cryogels (482 kPa), accompanied by an improved $\varepsilon_f$ (540%; Supplementary Movie 2). Furthermore, we observed that the order of $E$ for the suppressed cryogels not only breaks the mechanical limit of stiff cryogels but also makes it possible to well match them with those of ultrasoft tissues (1–10 kPa; such as neuron cell and skin[17–19]), remarkably expanding the potential applications of green cryogels. This is confirmed by the phenomenon that

the suppressed cryogels are capable of closely contacting with skin (inset of Fig. 3c) and freely bending with the elbow (Supplementary Fig. 5a), which are ascribed to its ultrasoftness and self-adhesiveness (Fig. 3h). The Mooney–Rivlin profiles of tensile data for suppressed cryogels and cryogels given in Supplementary Fig. 5b provide a deep insight of the tensile behaviors. A plateau region at large deformation followed by strain hardening under extreme elongation is observed only in the suppressed cryogels ($C_{CaCl2} \geq 2$ M), indicating ideal rubber-like behaviors[31]. Furthermore, their excellent resilience confirms this rubber-like nature, even at a large strain of 300%, wherein the suppressed cryogels ($C_{CaCl2}$ = 3 M) exhibit no hysteresis or fatigue-free behaviors even after 100 cycles (Supplementary Figs. 5c, d). These unique tensile behaviors of suppressed cryogels arise from their particular amorphous and H-bonding cross-linked network, easily

changing the polymer conformation because of the amorphous chains and the sliding between polymer chains with the breakage of H-bonds under external load. More importantly, the mechanical parameters of PVA cryogels could be extensively tuned by altering the distance between the adjacent chains ($d$), mainly including the factor of $CaCl_2$ concentration ($C_{CaCl2}$), PVA concentration of the precursor solution ($C_{PVA}$), cryogenic temperature ($T_c$), and thawing temperature. Here, it should be pointed out that the primary purpose of this work is the freezing temperature control for water molecules ($T_f$) via altering $CaCl_2$ concentration. However, in order to obtain a detailed data rule, other ultralow cryogenic temperatures ($T_c = -50\,°C$, $-80\,°C$) were also tried using a cryogenic medical refrigerator (MDF-U5386S, PHCbi). First, both $C_{CaCl2}$ and $T_c$ predominately influence the aggregation of polymer chains in cryogenic process. With an increase in $C_{CaCl2}$ or $T_c$, the crystallization of water molecules becomes weak (Fig. 3e), consequently weakening the approach of polymer chains and creating a large $d$. Second, the $C_{PVA}$ determines the initial $d$ in the precursor solution, and a low $C_{PVA}$ favors a large $d$. Thus, close packing of the polymer chains or a small $d$ can be achieved by decreasing $C_{CaCl2}$, lowering $T_c$, or increasing $C_{PVA}$, promoting $E$ of hydrogels (Fig. 3c and Supplementary Fig. 6a–c) to various extents. Among these factors, changing $C_{CaCl2}$ is more effective in adjusting the $E$ value. Notably, although a low $T_c$ favorably enhances $E$, an ultralow $T_c$ (e.g., $-80\,°C$) may induce a considerably fast freezing, wherein polymer chains have insufficient time to rearrange, resulting in a low $E$. Third, a high thawing temperature facilitates the mobility of polymer chains upon thawing, resulting in a large $d$ and low $E$ (Supplementary Fig. 6e). Because the cryogenic temperature of $-20\,°C$ and thawing temperature of $27\,°C$ are easily obtained[21,27], they have been widely adopted. Through varying these factors, the $E$ of PVA cryogels ranges from 5 to 123 kPa (Fig. 3d), almost covering all ranges of $E$ for the soft tissues (1–100 kPa)[17–19]. Moreover, different factors can cooperatively alter $d$. For example, the suppressed cryogels with $C_{PVA} = 14$ wt%, $C_{CaCl2} = 4$ M, and $T_c = -20\,°C$ are unstable at ambient temperature, due to that a large $d$ is triggered by a considerable level of antifreezing for water with a high $CaCl_2$ concentration. However, considering the reducing $d$ effect of high $C_{PVA}$, with $C_{PVA}$ increasing to 20 wt%, the suppressed cryogels with $C_{CaCl2} = 4$ M and $T_c = -20\,°C$ become well stable (Supplementary Fig. 6b). In comparison with existing PVA hydrogels fabricated using various methods[15,30,32], the suppressed cryogels prepared using the SFT strategy exhibit the lowest Young's modulus (5–10 kPa) along with superior extendibility (-600%), endowing the suppressed cryogels with flexible nature (Fig. 3g).

Notably, the generation of free –OH groups in H-bonding crosslinked networks also endows suppressed cryogels with self-adhesiveness to a wide range of substrates (such as glass, plastic, steel, wood, and pigskin) and even to polytetrafluoroethylene that is usually used as a nonstick coating (Supplementary Fig. 7a). The lap shear strength of PVA cryogels was characterized via sandwiching a hydrogel (thickness of 1 mm) within two polyamide films. Similar to mechanical variations, we observe that the adhesion strength (Fig. 3h, i) strongly depends on the factors influencing the distance between the adjacent chains ($d$). Thus, as the $d$ increases, the adhesion strength first increases and then decreases, reaching a maximum value at the critical $d$ ($d_c$). This event arises from the fact that (Supplementary Fig. 8), with $d < d_c$ (a small $d$ from high $C_{PVA}$ or low $C_{CaCl2}$), most of the –OH groups tend to form H-bonds in the self-hydrogel (as confirmed by Fig. 2e, f), generating a strong matrix. However, the interface between hydrogel matrix and substrates lacks free –OH groups to interact with each other, thereby resulting in adhesive failure during lap shear test (Fig. 3i and Supplementary Fig. 8c). In case of $d > d_c$ (a large $d$ from low $C_{PVA}$ or high $C_{CaCl2}$), most of the –OH groups are freely distributed and a small number of H-bonds make the hydrogels considerably weak. Although a strong interface could be generated between hydrogel and substrate due to a sufficient number of –OH

groups, the matrix failure preferentially occurs, inducing cohesive failure during the lap shear test (Supplementary Fig. 8a). Notably, in the critical $d = d_c$ (critical $C_{PVA}$ or $C_{CaCl2}$), combinational free –OH and H-bonding –OH groups endow both stable hydrogel matrix and strong interface between hydrogel and substrate, producing the maximum adhesion strength (Supplementary Fig. 8b). Thus, with a pre-determined $C_{PVA}$, the adhesion strength first increases and then decreases with increasing $C_{CaCl2}$, whereas with a predetermined $C_{CaCl2}$, a similar variation trend is observed with increasing $C_{CaCl2}$. Notably, with an increase in $C_{CaCl2}$, the critical $C_{PVA}$ corresponding to the strongest self-adhesion is gradually increased (Fig. 3h) due to its cooperative role in altering $d$. Furthermore, our investigations revealed that the freezing and thawing temperatures influence the adhesion strength to some extent by altering the $d$ (Supplementary Fig. 6d, f). Thus, using an SFT strategy, the suppressed cryogels with ultraflexibility are demonstrated to exhibit excellent self-adhesiveness, endowing them with conformal contact to soft tissues such as skin and repeatable adhesive ability, even on curvilinear surfaces, upon irregular motion (Supplementary Movie 3). Moreover, they could serve as an antiscratch, transparent, and conductive coating on complex 3D-shaped materials (Supplementary Fig. 7b).

Another striking property of suppressed cryogels is their instantaneous self-healing capacity, even with low concentrations of PVA (-14 wt%), ensuring antiscratch ability and prolonging their application lifetime. For conventional PVA cryogels (Supplementary Fig. 9b), the semi-crystalline structure severely limits the mobility of the polymer chain, which could not diffuse across cutting surfaces. Moreover, most of the –OH groups have already formed H-bonds due to the small distance between adjacent chains, and the cutting interface lacks free –OH groups to form new H-bonds. Therefore, according to the literature, only those stiff PVA cryogels with a high polymer concentration (30–35 wt%) could repair themselves under a long healing time[20]. However, the suppressed cryogels proposed in the current research overcome these limitations. A particular amorphous structure with a high content of free –OH groups allow the suppressed cryogels to easily diffuse polymer chains and form H-bonds across the fresh surfaces, revealing their effective self-healing nature (Supplementary Fig. 9a). Moreover, the relatively weak cross-linking structure facilitates the reprocessing ability of suppressed cryogels as well, aiding their remolding into any desired shapes (Supplementary Fig. 10).

Overall, the emergence of multifunctionality (ultrasoftness, high transparency, self-adhesiveness, instantaneous self-healing, and reprocessing ability) for cryogels with a very simple recipe is primarily attributed to the unique homogenous hierarchical structures, amorphous polymer networks, and coexistence of free/H-bonding –OH groups. Notably, $CaCl_2$ aids in adding some additional properties to the hydrogel matrix. First, the water-retention capacity of the suppressed cryogels becomes considerably better (Supplementary Fig. 11a) due to the ionic hydration of a salt[33], which is confirmed by its much lower steady-state water loss (-10 wt%) than that of cryogels (-74 wt%). Thus, the size of the suppressed cryogels remains stable for a long time; however, the size of the conventional freeze-thawed cryogels is considerably reduced (Supplementary Fig. 11c). Second, the multifunctionality of the suppressed cryogels could be retained, even under ultracold environments, due to the depression of the freezing temperature of the water, particularly for its unusual freezing-tolerant transparency ($-196\,°C$; Supplementary Fig. 11b and Supplementary Movie 1). This finding indicates that both characters guarantee the long-term application of suppressed cryogels in harsh environments. Finally, compared with the narrow spacing between adjacent chains anchored by crystalline cross-linked points of cryogels, the amorphous structure endows the suppressed cryogels with a large distance between the adjacent chains, serving as a rapid transportation channel of ions[34–36] and remarkably improving the ionic conductivity

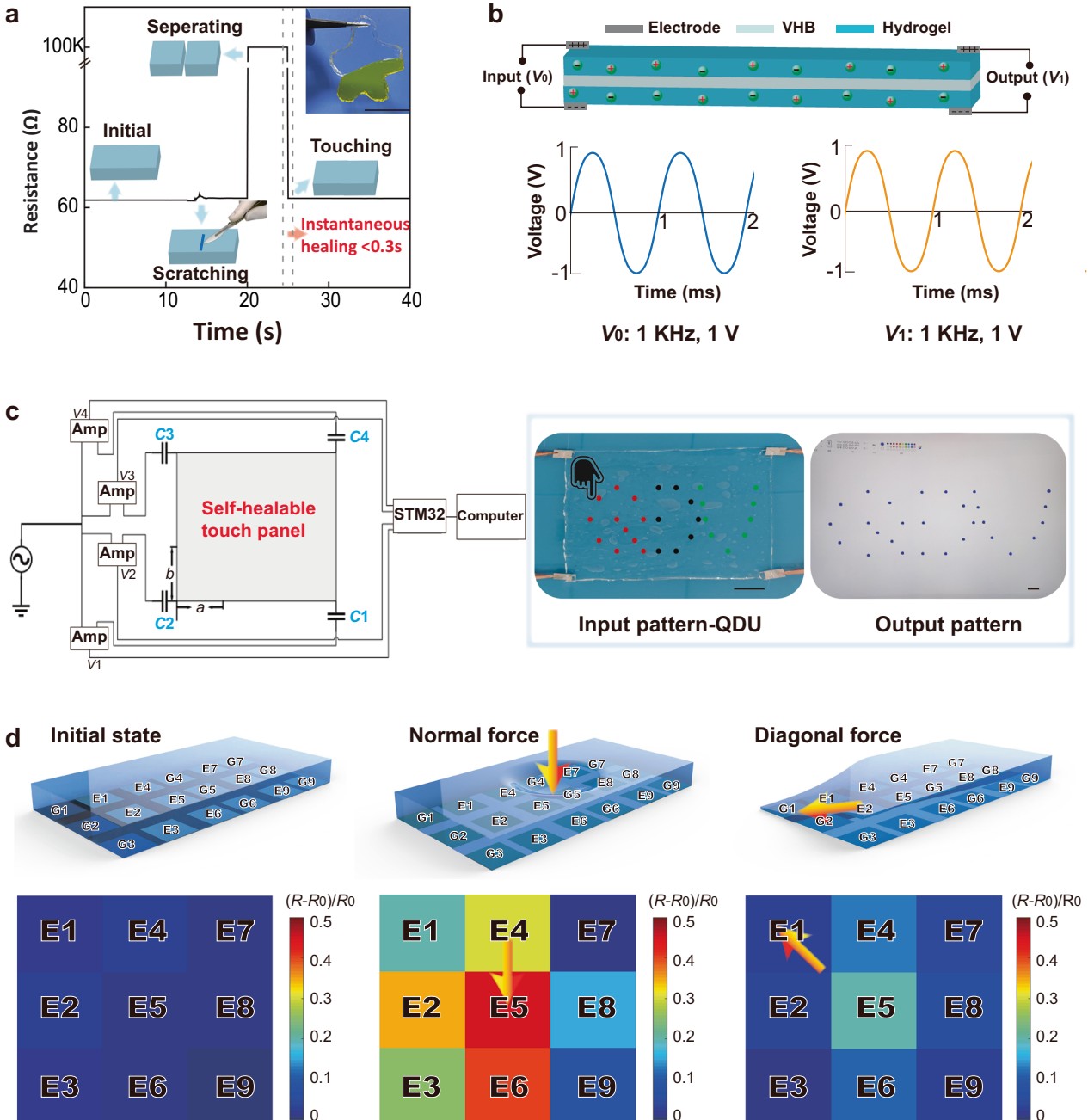

**Fig. 4 | Sensory applications based on the suppressed cryogels. a** Resistance changes in suppressed cryogels under an external field. **b** Structural design (top) and input/output signal waveforms (both frequency and amplitude remain unchanged) (bottom) for suppressed cryogels in the application of signal transmission. **c** The design principle of a self-healing touch panel (left) using suppressed cryogels and its pattern demonstration (right). The normalized distance of *a* and *b* are introduced when the position of the suppressed cryogel is converted to the computer. *V*1, *V*2, *V*3, and *V*4 are the voltage after the current is amplified by four operational amplifiers. *C*1, *C*2, *C*3, and *C*4 are the electric double-layer capacitance between the platinum sheet and the suppressed cryogel. **d** Cryogels in the application of pressure sensor, respectively, revealing the deformation responses for hydrogels (top) and resistance changes (bottom) under no force, normal force, and combined tangential and normal forces. (E1–E9 represent the four working electrodes, G1–G9 represent the ground electrodes, $R_0$ represents the initial resistance of the sensor, and $R$ represents the resistance of the sensor during the stress test.) Scale bars: 30 mm in (**a**, **c**).

(Supplementary Fig. 12), thereby enlarging their application range as flexible electronic materials (Fig. 4).

Notably, according to the mechanism of the SFT approach, the key part to this strategy is the choice of salt and its pre-incorporation into the precursor solution to suppress the ice size upon freezing. For instance, salting-out[16] NaCl or KCl without antifreezing ability did not work and only boosted the polymer aggregation. The already formed cryogels soaking with $CaCl_2$ solution for a long time remained similar nature with that of common cryogels.

## Sensory applications of the suppressed cryogels

Under the action of multi-scale structures and the nature of salts, such a suppressed cryogel with a very simple recipe yet appealing multifunctionality holds important potential in sensory applications. Based on the real-time resistance change of the hydrogel (Fig. 4a), when the suppressed cryogel was scratched by a knife, only a slight change was observed in its resistance, indicating its excellent anti-scratch properties. This finding was confirmed to be related to the applied stress during the cutting process (Supplementary Fig. 13). Furthermore, the

cutting-suppressed cryogels ($C_{CaCl2}$ = 3 M) could immediately recover their ionic conductivity when two separating surfaces are brought into contact, maintaining stability after releasing the holding load and exhibiting instantaneous self-healing property (Supplementary Movie 4). However, this observation was not observed for cryogels soaked in the same $CaCl_2$ concentration for sufficient time.

These properties make suppressed cryogels ideal candidates in sensory applications. First, the suppressed cryogels can be used as electrolytes in the fabrication of artificial nerve fibers with pure platinum sheets as electrodes and VHB tapes as dielectric layers, where one end is the input port and the other end is the output port (the top of Fig. 4b). The function signal generator that sends out a sinusoidal voltage signal with a frequency of 1 kHz and an amplitude of 1 V could be connected to the input port of the artificial nerve fiber, after which an electronic oscilloscope is used to display the voltage signal curves at the input and the output port (the bottom of Fig. 4b). Furthermore, according to the comparative data, the artificial nerve fibers exhibit excellent information transmission performances. More importantly, the artificial nerve fibers are self-healing, stretchable (Supplementary Movie 5), and can work in ultralow temperature environments (Supplementary Movie 6). Simultaneously, the artificial nerve fiber could maintain good information transmission performance when stretched in a low-temperature environment (Supplementary Fig. 14). Second, the hydrogel can be applied in fabricating self-healing touch panels (Fig. 4c). The suppressed cryogels are combined with an AC power supply, operational amplifier, platinum electrode, STM32, and computer. Thus, the computer software can be controlled via the hydrogel touch panel. As shown in Fig. 4c, when the touch panel controls the drawing software, a QDU pattern input on the touch panel can be accurately displayed in the drawing software (see the "Methods" section and Supplementary Figs. 15, 16). Third, the suppressed cryogels can be fabricated into a pressure sensor (Fig. 4d). When operating the pressure sensor based on the suppressed cryogels with an AC voltage of 0.6 V amplitude and a frequency of 10 kHz (Supplementary Figs. 17, 18), we found that when the pressure ($P$) is <10 kPa, the sensitivity of the pressure sensor is 101.3 MPa$^{-1}$, and when 10 kPa <$P$< 100 kPa, the sensitivity is 0.913 MPa$^{-1}$ (Supplementary Fig. 19), with the response time of the pressure sensor being as fast as 18 ms (Supplementary Fig. 20a), which is comparable to other works[37–40]. Simultaneously, the pressure sensor has excellent stability and repeatability (Supplementary Fig. 20b). Based on these findings, a pressure sensor that can discern the distribution and direction of force ($F$) was further developed, utilizing a pressure sensor array (see the "Methods" section and Fig. 4d). As shown in the left figure of Fig. 4d, the pressure sensor array comprised suppressed cryogels, nine pressure-sensitive electrodes, and nine ground electrodes, including one pressure-sensitive electrode and one ground electrode (such as E1 and G1) to form a sensing area (Supplementary Fig. 21a) and a total of nine sensing areas (E1, E2, E3, E4, E5, E6, E7, E8, and E9) to create a 3*3 electrode array (Supplementary Fig. 21b). When a force is applied, its distribution and direction can be determined using the differential signal of the sensor. For example, with a normal force applied in the E5 region, the resistance change of E5 is the largest, the resistance changes of E6, E4, E2, E1, E3, and E8 in turn decrease, and the resistance changes of the other two regions are smaller (the middle figure of Fig. 4d). When a diagonal force in the direction of E5 to E1 is applied, the resistance change of E1 is the largest, the resistance changes of E2, E4, E3, and E5 decrease in turn, and the resistance changes of the other four regions are smaller (the right figure of Fig. 4d).

We believe that our strategy provides both material and conceptual breakthroughs for multifunctional cryogels with considerably simple recipes, offering opportunities in the fields of skin-attachable electronic devices, robotics (particularly in low-temperature environments), and prosthetics.

## Discussion

The multifunctional cryogel is achieved through creating an amorphous cross-linked network with the coexistence of free and H-bonding hydroxyl groups. These unique multi-scale structures are derived from a distinctive microstructure evolution upon freezing, that is, instead of enhancing the polymer-aggregation-triggered polymer crystallization, a considerably weak approach of polymer chains is induced via depressing the growth of ice crystals at the initial freezing gelation stage. Thus, an entirely amorphous yet relatively homogenous polymer network is created in freeze-thawed hydrogels, causing a large distance between adjacent chains to make free −OH groups available. In this way, the suppressed cryogels with a simple recipe break the sole semi-crystalline structure and restricted performance of cryogels, representing the integrated properties (such as tissue-like ultrasoftness, transparency, self-adhesiveness, instantaneous self-healing, superior ionic conductivity, and antifreezing). Overall, our strategy provides both material and conceptual breakthroughs for developing multifunctional cryogels, considerably increasing the range of applications for cryogels, particularly in advanced fields, such as artificial nerve fibers and self-healing touch panels, as demonstrated in this research.

## Methods

### Synthesis of suppressed cryogels

The common polyvinyl alcohol (PVA) cryogels were prepared following FT procedures. The PVA powder (degree of polymerization 1700, degree of alcoholysis of 99%, Sinopharm Chemical Reagent Co., Ltd.) was pre-swollen in water overnight, and the mixture solution was stirred at 95 °C for 2 h at 300 rpm. After ultrasonic defoaming, the obtained solution was poured into a glass mold with a silicon separator (2 mm). The mold was freezing at −20 °C for 12 h and thawing at 27 °C for 12 h, which was repeated for 3 cycles. The suppressed cryogels were synthesized by the following procedures. The $CaCl_2$ (AR, Sinopharm Chemical Reagent Co., Ltd.) aqueous solution was first prepared by stirring at 25 °C for 30 min. Then a certain amount of PVA powder was added and pre-swollen overnight. After that, the following procedures of making a mixture solution and suppressed cryogels are totally the same as those of common cryogels. In this investigation, the influence of $CaCl_2$ concentrations (1, 2, 3, 4, and 5 M), PVA concentrations (12, 14, 16, 18, and 20 wt%), cryogenic temperature (−20, −50, and −80 °C), and thawing temperature (4 and 27 °C) on the ultimate properties of PVA hydrogels were systematically investigated.

### Multi-scale structure characterizations

The small-angle X-ray scattering pattern (SAXS) and the wide-angle X-ray scattering pattern (WAXS, 0°–18°) were collected on the beamline BL16B1, which is located at Shanghai Synchrotron Radiation Facility. The wavelength of $\lambda$ = 1.24 Å and a detector of MAR CCD are used. The resolution is 2048 × 2048 pixels, and the pixel size is 172 × 172 µm. The exposure time of both the SAXS and WAXS is 40 s, and the sample-to-detector distance is 2036.4 mm. Further, the WAXS pattern (18°–40°) was collected on the beamline BL14B. The wavelength of $\lambda$ = 1.24 Å and a detector of MAR CCD are used. The resolution is 3072 × 3072 pixels, and the pixel size is 73 × 73 µm. The exposure time is 40 s, and the sample-to-detector distance is 361 mm. Using Fit 2d software, every photo was corrected for background and air scattering and then integrated them into one-dimensional curves.

Fourier transform infrared spectroscopy (FTIR) was used to collect the FTIR spectra of the hydrogels in a wavenumber range of 4000–400 cm$^{-1}$. An attenuated total reflection (ATR) mode was adopted. At the resolution of 2 cm$^{-1}$, the spectra were obtained by 32 scans. Using OriginPro software, the peak located in the range of 3850–2600 cm$^{-1}$ was fitted into two absorption peaks of 3370 and 3210 cm$^{-1}$ based on Gaussian−LorenCross functions.

## Tensile measurements

Tensile properties were collected using an electric universal tensile machine. The hydrogel was cut into dumbbell-shaped splines ($35 \times 2 \times 2$ mm, length × width × thickness) and underwent a uniaxial stretching with a velocity of 50 mm/min. For cyclic tensile tests, the hydrogel was stretched to various strains (100%, 200%, 300%, 400%, and 500%) at 50 mm/min, followed by unloading to the initial location. To further obtain its fatigue resistance, the splines were loaded to a strain of 300% and then unloaded to the initial location, which was repeated for 100 cycles. Lap shear measurements were performed to evaluate the adhesion strength of the hydrogels. The hydrogel was sandwiched between two sheets of substrates (polyamide film, PA) and then subjected to uniaxial tensile tests. The test sample was prepared as follows. The precursor solution was poured between two PA films with a silicon separator ($20 \times 20 \times 1$ mm, length × width × thickness), which was sandwiched by two glass plates and fixed by the clamps. The lap shear test was undergone at 20 mm/min, and each sample is repeated with at least six parallel samples.

## The fabrication of artificial nerve fibers and their signal transmission

A dielectric elastomer (3 M VHB 5608) was placed in the middle of two parallel hydrogels (length 200 mm, width 10 mm, height 2 mm), and four pure platinum sheets were connected to the hydrogel terminals (as shown in Fig. 4b). A function signal generator (SP-F40) connected to the input port of artificial nerve fiber to generate a sinusoidal signal with an amplitude of 1 V and a frequency of 1 kHz. Both input and output signals are displayed by an electronic oscilloscope (UTD2102CEX).

## The fabrication of self-healing touch panel and its pattern demonstration

The self-healing touch panel consists of a piece of hydrogel (length 200 mm, width 15 mm, thickness 2 mm), function signal generator (SP-F40), operational amplifier (OPA177), microcontroller (STM32-F103-V2), pure platinum sheet and a computer.

As shown in Fig. 4c, a sine signal with an amplitude of 0.6 V and a frequency of 30 kHz is emitted by the function signal generator. The sine signal is connected to the hydrogel through a pure platinum sheet after passing through the operational amplifier. When the hand contacts the hydrogel, the simulation of the entire module is shown in Supplementary Fig. 15. After that, the current signal is converted into a voltage signal by the operational amplifier and then sent to the STM32 for processing, and finally displayed on the computer. (Fig. 4c)

In order to make the conversion of position between the hydrogel and the computer more accurate, two normalized distances $a$ and $b$ are introduced. In combination with Supplementary Fig. 15, the position can be calculated according to Eqs. (1) and (2).

$$a = \frac{V2 + V3}{V1 + V2 + V3 + V4} \tag{1}$$

$$b = \frac{V3 + V4}{V1 + V2 + V3 + V4} \tag{2}$$

Among them, $V1$, $V2$, $V3$, $V4$ are the voltages after the current is amplified by the four operational amplifiers, respectively.

To verify the accuracy of the obtained formula, as shown in Supplementary Fig. 16, four points were selected on the hydrogel for testing.

## The fabrication of pressure sensor and pressure sensor array

Supplementary Fig. 21a shows the manufacturing process of the pressure sensor. First, the silver electrodes were printed on the flexible PET substrate by 3D printing. Secondly, the VHB with a thickness of 0.1 mm in a specific shape is placed on the PET substrate. Then a pure platinum sheet (length 5 mm, width 5 mm, height 0.1 mm) was placed on the corresponding position of the silver electrode. Finally, the hydrogel was placed on it. As shown in Supplementary Fig. 21b, the pressure sensor array was fabricated in the same way.

## Data availability

The data supporting the findings of this study are available within this article and its Supplementary Information. All data is available from the corresponding author upon request. Source data are provided with this paper.

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

## Acknowledgements

The current work was financially supported by the National Natural Science Foundation of China (Grant Nos. 22273042 and 52003134) to L.L.W. and M.L.J., Natural Science Foundation of Shandong Province, China (Grant No. ZR2023YQ042) to L.L.W., Program for Changjiang Scholars and Innovative Research Team in University (Grant No. IRT_14R30) to Y.Z.X., and Taishan Scholar Program of Shandong Province (Grant Nos. tspd20181208 and tsqn201909099) to Y.Z.X. and M.L.J. We also thank the beam time on BL14B1 and BL16B1 in SSRF of China.

## Author contributions

L.L.W. and M.L.J. conceived the idea and designed the project. X.S.Z., C.Z.X., and H.W.Y. performed all the experiments. X.S.Z., C.Z.X., and Y.W. performed the characterization. H.W.Y. performed the sensing applications. A.P.F., H.L., and J.Y.L. performed the simulation. X.D., S.Z., J.H.N., M.G., J.W., J. P. Y., S. Z. G., and Y.Z.X. helped to analyze the data. X.S.Z., L.L.W., and M.L.J. wrote and revised the paper. L.L.W. and M.L.J. supervised the project.

## Competing interests

The authors declare no competing interests.
