## [Peer Review File · Nature Communications]

Skin-like cryogel electronics from suppressed-freezing tuned polymer amorphizationREVIEWER COMMENTS

Reviewer #1 (Remarks to the Author):

AS mentioned in Introduction, anti-freezing agents such as ethylene glycol have already been investigated for hydrogels. Ethylene glycol or other polyols are biocompatible although the authors of this manuscript argued about their bio toxicity. This work is not novel enough for this journal.

Reviewer #2 (Remarks to the Author):

The authors propose an unusual strategy of precisely reducing the size of ice crystals in the cryogenic process of cryogels, which achieves the amorphization of the polymer network and the coexistence of free and H-bonded hydroxyl groups. The idea is very interesting, and the novel hierarchical structures undoubtedly make significant changes in the multiple properties of cryogels. The results in this manuscript will facilitate many advanced applications for green cryogels. Some suggestions and questions are given in the following.

1. The author describes the formation principle of conventional cryogels and the proposed suppressed cryogels according to the phase diagram of the PVA/CaCl₂/H₂O solution in Figure 1a. I wonder what will happen for cryogels based on fabrication parameters in the purple region of the diagram.
2. The newly suppressed cryogels represent many interesting performances, particularly for their transparency even in liquid nitrogen. The author said that the hydrogels become rigid but transparent in liquid nitrogen. How to interpret this phenomenon from a molecular level?
3. The key to suppressed freezing strategy is the regulation of ice by an anti-freezing salt, which determines the subsequent formation of multiscale microstructures and the ultimate multifunctionality for cryogels. Did the author try other anti-freezing agents (e.g., glycerin or LiCl), to confirm the versatility of this method?
4. The multi-scale structures of cryogels are well characterized by SAXS, WAXS, and FTIR in this manuscript. The authors are suggested to compare the difference of morphology for hydrogels, such as the microporous structure according to SEM.
5. The long-term stability of hydrogels is very important for their practical applications.

Although the author has confirmed that the suppressed cryogels keep well water retention ability, the low water loss still exists according to Figure S10a. The authors are suggested to make further improvements in long-term stability to satisfy the stringent requirement of advanced applications.

6. The authors claim to have fabricated a capacitive pressure sensor using the proposed suppressed cryogel, but there is less description of the sensor's sustained force stability, and needs to be supplemented.

7. The authors' work is interesting and relevant for artificial nerve fibers, especially for studies at low temperatures. I'm curious about the changes in the properties of artificial nerve fibers when they are stretched at low temperatures.

Reviewer #3 (Remarks to the Author):

Comments:

Recommendation for publication in Nature communications after major revision.

The manuscript entitled "Skin-like cryogel electronics from suppressed-freezing tuned polymer amorphization" submitted by Lili Wang and coworkers presented a simple strategy to achieve multifunctionality in cryogels by total amorphization of polymer. The suppressed cryogels showed tissue-like softness, transparency, self-adhesiveness, instantaneous self-healing, superior ionic conductivity, and anti-freezing. While the material design of this work is quite interesting, there are several unsupported claims for the multifunctionality of suppressed cryogels throughout the figures and text.

1) The detailed mechanisms of how suppressed cryogels accomplish self-adhesion, self-healing, and high ionic conductivity could be the key points of this work; nevertheless, there are no sufficient experiments and discussions to explain the origin of their multifunctionality.

2) Additionally, the authors claim "superior ionic conductivity". There is no experiment done to test electrochemical properties (e.g., impedance spectroscopy), so this claim is not supported by any data. There was only one resistance response result without any comparison to the state of art solid-state electrolyte. Because of these two reasons, the

impact of this work may be limited. Overall, I recommend the addition of more experiments and appropriate discussion for the aforementioned issues. Here are some detailed comments:

1. Fig. 1(b) and (c) are not mentioned in the text. Please mention these items on page 6.
2. The authors need to provide the chemical mechanism or molecular interaction behind CaCl₂'s function as an anti-freezing agent.
3. Page 11, line 21; "With an increase in C_{CaCl2} or a decrease of T_c, the crystallization of water molecules becomes weak, consequently weakening the approach of polymers and creating a large d.". Page 4, line 18; "instead of changing the cryogenic temperature (T_c), this study successfully tackled this issue by ingeniously lowering the freezing temperature (T_f) of water molecules, assisted by an anti-freezing salt (CaCl₂)". What kind of temperature control (T_c or T_f) is the primary purpose of this work?
4. Fig. S1, cryogel solution with CaCl₂ at -20°C seems liquid phase. In this system, salt would be hydrated with water molecules, and would generate ion-dipole interaction between ions and the hydroxyl group of PVA. Fig. 1 is missing the representation of molecular interactions between ions and water.
5. Fig. 4, sensory applications experiment, 1 kHz biphasic pulse of 1V amplitude was used to operate the suppressed cryogels-based sensors. How did the authors determine the pulse parameters for the input source? Is there any concern regarding electrolysis or redox reactions?
6. Please double check some typos.

Response to Referees (NCOMMS-23-03385A)

Reviewer #1:

Comments

AS mentioned in Introduction, anti-freezing agents such as ethylene glycol have already been investigated for hydrogels. Ethylene glycol or other polyols are biocompatible although the authors of this manuscript argued about their bio toxicity. This work is not novel enough for this journal.

Response: Many thanks for reviewer's comments. We totally agree with the reviewer that many kinds of anti-freezing agents, including anti-freezing salts (CaCl_2 and LiCl), organic solvents (ethylene glycol, dimethyl sulfoxide and glycerol) and zwitterionic penetrants (betaine and proline) have already been investigated for hydrogels. These investigations mainly use the intrinsic anti-freezing nature of these agents to improve the freezing tolerance of hydrogels.

However, the primary purpose of our work is not to improve the freezing tolerance for hydrogels or proposing an anti-freezing agent of hydrogel. The focus of our work is to **break the sole situation of semi-crystalline structures induced single performance for freeze-thawed cryogels** (Fig. R1b-i). Based on the mechanism of freezing gelation (Fig. R1b-ii), the growth of ice expels the polymer chains (Fig. R1a) and determines the ultimate multi-scale structures/multiple properties of cryogels. **Here, we innovatively select an anti-freezing agent to suppress the growth of ice, which induces an unusually weak yet homogenous approach of polymer chains during freezing gelation and successfully achieves the distinctive amorphous multi-scale structures with both H-bonding and free hydroxyl groups (Fig. R1b-iii)**. As confirmed by our work, the newly integrated properties (tissue-like ultra-softness (Young's modulus, 4~10 kPa) yet stretchability, high transparency (~92%), self-adhesion and instantaneous self-healing) are ascribed to this particular amorphous multi-scale structures of suppressed cryogels, which are impossible for conventional semi-crystalline cryogels. Of course, the incorporation of anti-freezing salt also endows its intrinsic nature to hydrogels, including ionic conductivity, anti-freezing and water-retention ability. With these appealing traits together, the capacity of using single cryogels are first demonstrated in a variety of advanced sensory applications. Therefore, the introduction of anti-freezing agents in our work is mainly to regulate the evolution of polymer chains during freezing gelation and the ultimate multi-scale structures, rather than

just improving freezing tolerance of hydrogel.

Fig. R1 **a** The polymer evolution during freezing gelation (i) and the corresponding structures at multi-scales (ii–iii) for cryogels. **b** The general freezing gelation mechanism for freeze-thawed hydrogels (ii), and the specific condition for conventional cryogels (i) and the new suppressed cryogels with anti-freezing salt (iii).

In the following, please allow us to describe the novelty of our work from various aspects.

1. Background

The green cryogels have attracted risen attention in recent years owing to their remarkable advantages of nontoxicity, simple fabrication without harsh synthetic condition, biocompatibility and their similarity to biological tissues. However, **polymer aggregation-induced semi-crystalline structures** make cryogels suffer from extremely restricted functions and sharing the integrated properties (e.g., flexibility, transparency, self-adhesiveness and self-healing) is almost impossible. This situation remains since their discovery due to hardly controlled **freezing fabrication technology**. Therefore, there is an urgent need for the development of the new multi-scale microstructures via a distinctive forming mechanism in cryogels to trigger their structural merits to maximum extent, which would be promising to achieve multifunctionality and satisfy the stringent requirements of emerging applications, such as ionic skin, wearable sensors and optical devices.

2. Innovations

(1) Strategy

Based on the mechanism of freezing gelation, freezing-induced ice crystals result in phase separation of polymers, wherein the polymer crystallization with the formation of hydrogen bonds occurs in the polymer-rich region and these interactions remain intact after thawing, serving as physical cross-linking points (Fig. R1a). Accordingly, the phase separation of polymer chains and the ultimate semi-crystalline structure of cryogels are determined by the growth of ice during freezing gelation. Therefore, in our work, it is assumed that if **the ice crystals are suppressed, its expelling effect on polymer chains and the resultant phase separation (polymer-rich and polymer-poor phases) will be greatly weakened**, which is promising to produce a distinctive polymer chains evolution and microstructures/properties for cryogels. Here, in order to overcome the difficulty of precisely controlling the external cryogenic temperature and the rapid growth of ice, **an anti-freezing salt CaCl₂ is preferentially selected to precisely regulate the growth of ice**. As confirmed in our work, the incorporation of CaCl₂ into precursor solution of hydrogel could effectively suppress the growth of ice, and the freezing temperature (T_f) of precursor solution for hydrogels could be widely altered (0 ~ -58 °C) by varying CaCl₂ concentration (Fig. 1a in the manuscript).

(2) Multi-scale structures

In combination with SAXS/WAXS and FTIR, we confirmed that the strategy of **anti-freezing salt assisted suppressed freezing** generates **the amorphization of polymer network and simultaneously release part of free H-bonding hydroxyl groups for the first time** (Fig. 2 in the manuscript) due to large distance between adjacent chains. These homogeneously amorphous network with combinational H-bonding and free -OH groups for suppressed cryogels breaks the structural limit of semi-crystalline cryogels with just H-bonding -OH groups. Notably, the arrangement manner of the polymer chains can be adjusted widely via altering the concentration of CaCl₂.

(3) Multifunctionality

The amorphous network and the releasing of free -OH groups endow the simple-recipe cryogels with the **integrated properties** of tissue-like ultra-softness (Young's modulus of 4~10 kPa, two magnitude decrease) yet stretchability (~600%), high transparency (~92%, transparent even in liquid nitrogen), self-adhesion, instantaneous self-healing (<0.3 s) and remoldability (Fig. 3 in the manuscript), all of which are impossible in conventional semi-crystalline cryogels. The intrinsic nature of anti-freezing salt also adds extra properties to

hydrogels, including ionic conductivity, anti-freezing and water-retention ability. Notably, the amorphization of polymer in our work also facilitates the insertion and rapid transport of ions due to enlarged distance between adjacent chains, which also has been confirmed in other researches (Yang, C. P. *et al. Nature* 2021, **598**, 590).

(4) Sensing applications

With these appealing traits, the capacity of using single cryogels are demonstrated in a variety of advanced sensory applications, such as **artificial nerve fibers, transparent touch panel and pressure sensor** (Fig. 4 in the manuscript). Especially the artificial nerve fibers, which have excellent transmission efficiency under low-temperature stretching conditions (Supplementary Fig. 14). Thus our strategy provides both material and conceptual breakthroughs for developing new generation of multifunctional cryogels, which also offers new opportunity in the fields of skin-attachable electronic devices, robotics and prosthetics.

Other Responses

(1) The selection of anti-freezing agents in regulating polymer evolution during freezing gelation. As aforementioned, the anti-freezing agents in this work is to regulate the freezing behavior of precursor solution. Therefore, their incorporation should not influence the solubility of polymers in water, especially under high polymer concentration. Here, the anti-freezing salts with “salting-in” effect (facilitating the dissolving of polymer in water, Wu, S. W. *et al. Adv. Mater.* 2021, 33, 2007829) are preferentially adopted. It should be pointed out that several kinds of salts have been tried, including NaCl, KCl and LiCl (Fig. R2-1 in Response to Reviewer 2#). The results confirmed our proposed mechanism that only those effective anti-freezing agents with “salting-in” effect (CaCl₂ and LiCl) could fabricate suppressed cryogels with multifunctionality. Additionally, ethylene glycol and glycerin were also tried, but they would induce weak gelation of precursor solution after incorporation. Therefore, considering green nature, the solubility of polymer for precursor solution and their role in suppressing growth of ice during freezing gelation, CaCl₂ is confirmed as the best choice in PVA system. In order to make it clear, the choice principle of anti-freezing agents and their interaction with water molecules are further emphasized in the revised manuscript (Supplementary Fig. S1a, Description on page 5). Moreover, we agree with the reviewer that ethylene glycol or other polyols are biocompatible, and we have modified the previous description (Page 5) into in a more appropriate way.

(2) The correlation between multi-scale structures and multiple properties. The focus of our work is the newly amorphous multi-scale structures and multiple properties of

cryogels. In order to make the manuscript supportive and readable, the correlation between multi-scale structures and multiple properties and necessary experiments are further complemented, including the mechanism of self-adhesiveness (Supplementary Fig. 8, Description on page 13), self-healing (Supplementary Fig. 9, Description on page 14), highly ionic-conductivity (Supplementary Fig. 12, Description on page 15) and long-term stability (Supplementary Fig. 11c).

Reviewer #2:

Comments The authors propose an unusual strategy of precisely reducing the size of ice crystals in the cryogenic process of cryogels, which achieves the amorphization of the polymer network and the coexistence of free and H-bonded hydroxyl groups. The idea is very interesting, and the novel hierarchical structures undoubtedly make significant changes in the multiple properties of cryogels. The results in this manuscript will facilitate many advanced applications for green cryogels. Some suggestions and questions are given in the following.

2-1. The author describes the formation principle of conventional cryogels and the proposed suppressed cryogels according to the phase diagram of the PVA/CaCl₂/H₂O solution in Figure 1a. I wonder what will happen for cryogels based on fabrication parameters in the purple region of the diagram.

Response 2-1. Many thanks for reviewer's careful review. The cryogels fabricated based on purple region of phase diagram in Fig. 1a have been fabricated. With high concentration of CaCl₂ ($C_{\text{CaCl}_2} = 4 \text{ M}, 5 \text{ M}$), the precursor solution became opaque upon freezing (Supplementary Fig. 1c) due to formation of CaCl₂·6H₂O, but the resultant hydrogels after thawing became unstable and dissolved at ambient temperature. This instability may be due to the great suppression of water crystallization under large ΔT , which leads to a loosely cross-linked network and a large distance between the adjacent PVA chains. In order to make it clear, the description of ultimate hydrogels in this region is emphasized in the revised manuscript.

Modification: (Page 6) the precursor solution becomes opaque upon freezing (Supplementary Fig. 1b) due to formation of CaCl₂·6H₂O²², but the resultant hydrogels after thawing are unstable and dissolved at ambient temperature.

2-2. The newly suppressed cryogels represent many interesting performances, particularly for their transparency even in liquid nitrogen. The author said that the hydrogels become rigid but transparent in liquid nitrogen. How to interpret this phenomenon from a molecular level?

Response 2-2. Thanks a lot for reviewer's questions. The transparency of hydrogels in the liquid nitrogen is ascribed to the anti-freezing nature of CaCl₂. These ions could interact with water molecules to form bound state, which destroys the original hydrogen bonding network between free water molecules and thus hampers the formation of ice crystals upon cooling (Supplementary Fig. 1a). Therefore, the freezing-tolerant transparency arises from that with sufficient ions bound with water molecules, very small number of free water

transforms into ultra-small sized ice crystals. The greatly reduced mobility of polymer chains in liquid nitrogen makes the hydrogel rigid, and it could be rapidly converted back once the hydrogels were put back in the ambient condition. In order to make it clear, the schematic illustration of molecular interaction between CaCl_2 and water molecules is complemented in Supplementary Fig. S1a, and the mechanism discussion of freezing tolerance for hydrogels in liquid nitrogen is added in the revised manuscript.

Modification: (Page 10-11) The freezing-tolerant transparency arises from that with sufficient ions bound with water molecules, very small number of free water transforms into ultra-small sized ice crystals. The greatly reduced mobility of polymer chains in liquid nitrogen makes the hydrogel rigid, and it can be rapidly converted back once the hydrogels are put back in the ambient condition.

Supplementary Fig. S1 a The schematic illustration of molecular interaction between CaCl_2 and water molecules, representing anti-freezing mechanism.

2-3. The key to suppressed freezing strategy is the regulation of ice by an anti-freezing salt, which determines the subsequent formation of multiscale microstructures and the ultimate multifunctionality for cryogels. Did the author try other anti-freezing agents (e.g., glycerin or LiCl), to confirm the versatility of this method?

Response 2-3. We really appreciate the reviewer's constructive suggestions. Considering that the anti-freezing agents is to regulate the freezing behavior for precursor solution, their incorporation should not influence the solubility of polymers. Therefore, the anti-freezing salts with "salting-in" effect¹⁶ (facilitating the dissolving of polymer in water) are preferentially adopted. Several kinds of salts have been tried in the current research, including NaCl , KCl and LiCl (Fig. R2-1). The results also confirm our proposed mechanism that only those effective anti-freezing agents with "salting-in" effect (CaCl_2 and LiCl) could fabricate suppressed cryogels with multifunctionality. Considering the green nature of suppressed cryogels, the corrosive LiCl is not discussed in this research. Additionally, the glycerin the reviewer suggested and the ethylene glycol are also tried, but they will induce weak gelation of precursor solution. Therefore, considering both green

nature and their role in suppressed freezing strategy, CaCl_2 is the best choice in PVA system.

References

- 16 Wu, S. W. *et al.* Poly(vinyl alcohol) Hydrogels with Broad-Range Tunable Mechanical Properties via the Hofmeister Effect. *Adv. Mater.* **33**, 2007829 (2021).

Fig. R2-1 The appearance of PVA hydrogels by the incorporation of NaCl and KCl in the precursor solution. The stress-strain curves and the transparency of suppressed cryogels with LiCl as anti-freezing agents ($C_{\text{PVA}} = 17 \text{ wt}\%$).

2-4. The multi-scale structures of cryogels are well characterized by SAXS, WAXS, and FTIR in this manuscript. The authors are suggested to compare the difference of morphology for hydrogels, such as the microporous structure according to SEM.

Response 2-4. Thanks a lot for reviewer's suggestions. In fact, the morphology of suppressed cryogels derived from SEM have been tried in our investigation. The first step for SEM characterization of hydrogel is preparing freeze-dried sample. However, our obtained suppressed cryogels have superior anti-freezing ability (as confirmed in Fig. S11b), which leads to the failure of freezing during freezing dry process. As a result, the structure of freeze-dried sample collapsed, without keeping the initial volume (Fig. R2-2). Therefore, we are sorry that the SEM characterization was unavailable for suppressed cryogels.

Fig. R2-2 The appearance of freeze-dried suppressed cryogels.

2-5. The long-term stability of hydrogels is very important for their practical applications. Although the author has confirmed that the suppressed cryogels keep well water retention ability, the low water loss still exists according to Figure S10a. The authors are suggested to make further improvements in long-term stability to satisfy the stringent requirement of advanced applications.

Response 2-5. Many thanks for the reviewer's valuable suggestions. We totally agree with the reviewer that the long-term stability of hydrogels is very important for their practical applications. As shown in Supplementary Fig. 11a, for typical suppressed cryogels proposed in our work ($C_{CaCl_2} = 3\text{ M}$), only approximately 10 wt% weight loss occurred until reaching a stable state. According to our observation, the flexibility, transparency and self-adhesion remained during this process, whereas the conventional cryogels lost most of water (Supplementary Fig. 11c). Therefore, the long-term stability of suppressed cryogels could satisfy the stringent requirement of advanced applications. In order to make it clear, the size changes and the corresponding appearance of both hydrogels with five days are added in the revised Supplementary Information.

Modification: (Page 14) Therefore, the size of suppressed cryogels remained stable for a long time, but the conventional freeze-thawed cryogels largely shrank (Supplementary Fig. 11c).

Supplementary Fig. 11 c The size changes and the corresponding appearance of both

hydrogels with five days.

2-6. The authors claim to have fabricated a capacitive pressure sensor using the proposed suppressed cryogel, but there is less description of the sensor's sustained force stability, and needs to be supplemented.

Response 2-6. Many thanks for the reviewer's valuable suggestions. As the reviewer commented, we add the description of the sustained force stability of the pressure sensor to the article. As shown in Supplementary Fig. 20b, when the pressure sensor responds to dynamic pressure (0.76, 2, 10, 20, 50 KPa) under 0.6 V and 10 KHz alternating current, high stability and repeatability of resistance change are obtained.

Modification: (Page 18) Simultaneously, the pressure sensor has excellent stability and repeatability (Supplementary Fig. 20b).

Supplementary Fig. 20 b Stability and repeatability of the pressure sensor (200 cycles) under pressures of 0.76 KPa (black), 2 KPa (red), 10 KPa (blue), 20 KPa (green) and 50 KPa (purple).

2-7. The authors' work is interesting and relevant for artificial nerve fibers, especially for studies at low temperatures. I'm curious about the changes in the properties of artificial nerve fibers when they are stretched at low temperatures.

Response 2-7. Many thanks for the reviewer's question. As shown in Supplementary Fig. 14, in order to better observe the changes in the characteristics of the artificial nerve fibers when they are stretched at low temperature, the output signal of the artificial nerve fiber was tested when the tensile deformation was 100%, 200% and 300%, in the ambient temperature of $-45\text{ }^{\circ}\text{C}$, which proves that the artificial nerve fiber still has excellent stability when stretched at low temperature.

Modification: (Page 17) Simultaneously, the artificial nerve fiber could maintain good

information transmission performance when stretched in a low temperature environment (Supplementary Fig. 14).

Supplementary Fig. 14 Information transmission performance of artificial nerve fibers when stretched at low temperature. At $-45\text{ }^{\circ}\text{C}$, when the tensile deformation of the artificial nerve fibers is 100% (a), 200% (b) and 300% (c), the AC signals of the output signal V_1 at different amplitudes and frequencies are monitored.

Reviewer #3:

Comments The manuscript entitled "Skin-like cryogel electronics from suppressed-freezing tuned polymer amorphization" submitted by Lili Wang and coworkers presented a simple strategy to achieve multifunctionality in cryogels by total amorphization of polymer. The suppressed cryogels showed tissue-like softness, transparency, self-adhesiveness, instantaneous self-healing, superior ionic conductivity, and anti-freezing. While the material design of this work is quite interesting, there are several unsupported claims for the multifunctionality of suppressed cryogels throughout the figures and text.

3-1. The detailed mechanisms of how suppressed cryogels accomplish self-adhesion, self-healing, and high ionic conductivity could be the key points of this work; nevertheless, there are no sufficient experiments and discussions to explain the origin of their multifunctionality.

Response 3-1. Many thanks for the reviewer's useful suggestions. We are sorry that the mechanism of multifunctionality for suppressed cryogels was not described clearly due to insufficient discussion about relationship between distinctive multi-scale structures (Fig. 2) and multiple properties (Fig. 3-4). As confirmed by the combinational characterizations of SAXS/WAXS and FTIR data in Fig. 2, the suppressed cryogels proposed in the current work generate **a homogeneously amorphous network with combined H-bonding and free –OH groups**, breaking the structural limit of semi-crystalline cryogels with H-bonding dominated –OH groups.

First, the prerequisite of **self-adhesiveness** for hydrogels is the ability of forming attractive interaction between hydrogel and substrates. Although conventional PVA hydrogels with rich –OH groups are expected to show self-adhesiveness, they are totally nonstick. Because most of –OH groups have already formed hydrogen bonds between themselves and the system lacks free –OH groups to interact with substrates. In our work, the suppressed freezing strategy releases part of free –OH groups for the first time due to large distance between adjacent chains, making self-adhesion available for PVA cryogels. Moreover, as discussed in the manuscript, the **adhesive strength** of hydrogel is determined by both interface interaction and stability of the hydrogel matrix. With a small $d < d_c$ (high C_{PVA} or low C_{CaCl_2}), most of the –OH groups tended to form a large number of H-bonds in the self-hydrogel, generating a strong matrix. However, the interface between hydrogel matrix and substrates lacks free –OH groups to interact with each other and thus results in adhesive failure during lap shear test (Supplementary Fig. 8-iii). In the case of a large $d > d_c$ (low C_{PVA} or high C_{CaCl_2}), most of the –OH groups were freely distributed and a smaller

number of H-bonds made the hydrogels very weak. Although strong interface could be generated between hydrogel and substrate due sufficient number of –OH groups, the matrix failure preferentially occurred, inducing cohesive failure during the lap shear test (Supplementary Fig. 8-i). Notably, in the critical $d = d_c$ (critical C_{PVA} or C_{CaCl_2}), combined free –OH and H-bonded –OH groups endowed both stable hydrogel matrix and strong interface between hydrogel and substrate, producing the maximum adhesion strength (Supplementary Fig. 8-ii). In order to make clear, the schematic illustration of self-adhesiveness (Supplementary Fig. 8) and detailed discussion are complemented (Page 13).

Supplementary Fig. 8 The schematic illustration of self-adhesiveness and its variation of the distance between adjacent chains d .

Second, according to literatures²⁰, the key for an effective **self-healing** of hydrogels is to have high concentration of free –OH groups on polymer chains and sufficient polymer chain mobility to diffuse across the interface from two cut surfaces. For conventional PVA cryogels (Supplementary Fig. 9-ii), the semi-crystalline structure severely limits the mobility of polymer chain, and the polymer chains on both cutting surfaces could not diffuse from each other. Moreover, most of –OH groups have already formed H-bonds due to small distance between adjacent chains, and the cutting interface lacks free –OH groups to form new H-bonds. Therefore, according to literatures²⁰, only those PVA hydrogels with high polymer concentration (30~35 wt%) could self-healed. However, the suppressed

cryogels proposed in the current work overcame these structure limits. The particular amorphous structure with a high content of free –OH groups made the suppressed cryogels easier to diffuse polymer chains (as convinced by the diffusion of dye across the cutting surface in Supplementary Fig. 9-i) and form H-bonds across the fresh surfaces, showing effective self-healing nature. In order to make it clear, the schematic mechanism of self-healing, the observation of self-healing for conventional cryogels/suppressed cryogels with red dye (Supplementary Fig. 9), and the detailed discussion are complemented (Page 14).

Third, for ionic conductivity of hydrogels, it has been confirmed that the narrow spacing between polymer chains (d) due to crystalline domain limit the ion transportation, and expanding the d by creating amorphous network enable the insertion and rapid transport of ions³⁶. Therefore, compared with the narrow spacing induced by crystalline crosslinked points of cryogels, the amorphous structure with a large distance between adjacent chains for suppressed cryogels also allows rapid ionic transportation, which remarkably improves ionic conductivity (Supplementary Fig. 12) and enlarges their application range as electronic materials (Fig. 4).

References

- 20 Zhang, H. J., Xia, H. S. & Zhao, Y. Poly(vinyl alcohol) Hydrogel Can Autonomously Self-Heal. *Acs Macro Lett.* **1**, 1233-1236 (2012).
- 36 Yang, C. P. *et al.* Copper-coordinated cellulose ion conductors for solid-state batteries. *Nature* **598**, 590 (2021).

Modification: Paragraphs on Page 13-15.

3-2. Additionally, the authors claim “superior ionic conductivity”. There is no experiment

done to test electrochemical properties (e.g., impedance spectroscopy), so this claim is not supported by any data. There was only one resistance response result without any comparison to the state of art solid-state electrolyte. Because of these two reasons, the impact of this work may be limited. Overall, I recommend the addition of more experiments and appropriate discussion for the aforementioned issues.

Response 3-2. Many thanks for the reviewer’s careful review and suggestions. As shown in Supplementary Fig. 12, in order to better demonstrate the superior ionic conductivity, the electrochemical impedance spectroscopy test was carried out on the suppressed cryogels containing different concentrations of CaCl_2 , which was compared with the latest solid electrolyte¹⁻¹².

Modification: Supplementary Fig. 12.

Supplementary Fig. 12 The conductivity of suppressed cryogels. a-d Electrochemical

Impedance Spectroscopy of the suppressed cryogels. Using the electrochemical workstation, electrochemical impedance spectroscopy tests were performed on the cryogels containing different CaCl₂ concentrations. The frequency of impedance test is 0.01 Hz ~ 100 KHz, and the initial voltage is 10 mV. **e** The variation of conductivity for the cryogels with CaCl₂ concentrations. **f** Comparison of conductivity between suppressed cryogels and reported ionic hydrogels.

References

1. Mahabole, M. P., Bahir, M. M., Kalyankar, N. V. & Khairnar, R. S. Effect of incubation in simulated body fluid on dielectric and photoluminescence properties of nano-hydroxyapatite ceramic doped with strontium ions. *J. Biomed. Sci. Eng.* **5**, 396–405 (2012).
2. Wu, S. et al. Tough, anti-freezing and conductive ionic hydrogels. *NPG Asia Mater.* **14**, 65 (2022).
3. Xu, Y., Rong, Q., Zhao, T. & Liu, M. Anti-freezing multiphase gel materials: bioinspired design strategies and applications. *Giant* **2**, 100014 (2020).
4. Morelle, X. P. et al. Highly stretchable and tough hydrogels below water freezing temperature. *Adv. Mater.* **30**, 1801541 (2018).
5. Zhang, X. et al. Inorganic salts induce thermally reversible and anti-freezing cellulose hydrogels. *Angew. Chem. Int. Ed.* **58**, 7366–7370 (2019).
6. Ren, Y. et al. Ionic liquid-based click-ionogels. *Sci. Adv.* **5**, eaax0648 (2019).
7. Liu, Z. et al. Poly(ionic liquid) hydrogel-based anti-freezing ionic skin for a soft robotic gripper. *Mater. Horiz.* **7**, 919–927 (2020).
8. Rong, Q., Lei, W., Huang, J. & Liu, M. Low temperature tolerant organohydrogel electrolytes for flexible solid-state supercapacitors. *Adv. Energy Mater.* **8**, 1–7 (2018).
9. Yang, Y. et al. Anti-freezing, resilient and tough hydrogels for sensitive and large-range strain and pressure sensors. *Chem. Eng. J.* **403**, 126431 (2021).
10. Duan, S. et al. Tendon-inspired anti-freezing tough gels. *iScience* **24**, 102989 (2021).
11. Wu, S. et al. Rapid and scalable fabrication of ultra-stretchable, anti-freezing conductive gels by cononsolvency effect. *EcoMat* **3**, e12085 (2021).
12. Yao, B. et al. Hydrogel ionotronics with ultra-low impedance and high signal fidelity across broad frequency and temperature ranges. *Adv. Funct. Mater.* **32**, 2109506 (2022).

3-3. Fig. 1(b) and (c) are not mentioned in the text. Please mention these items on page 6.

Response 3-3. Great thanks for the reviewer's careful review. Fig. 1b and Fig. 1c have been added in the revised manuscript.

3-4. The authors need to provide the chemical mechanism or molecular interaction behind CaCl₂'s function as an anti-freezing agent.

Response 3-4. Thanks a lot for the reviewer's valuable suggestions. The schematic mechanism of CaCl_2 as an anti-freezing agent is complemented in Supplementary Fig. 1a, and the corresponding discussion is added in the revised manuscript.

Modification: (Page 5) First, the incorporation of CaCl_2 into water transforms some free water molecules (Supplementary Fig. 1a-i) into bound water molecules (Supplementary Fig. 1a-ii), inhibiting the original hydrogen bonding network between water molecules and the formation of ice crystals during cooling (Supplementary Fig. 1a-iii).

Supplementary Fig. 1 a The schematic illustration of molecular interaction between CaCl_2 and water molecules, representing anti-freezing mechanism.

3-5. Page 11, line 21; “With an increase in C_{CaCl_2} or a decrease of T_c , the crystallization of water molecules becomes weak, consequently weakening the approach of polymers and creating a large d.”. Page 4, line 18; “instead of changing the cryogenic temperature (T_c), this study successfully tackled this issue by ingeniously lowering the freezing temperature (T_f) of water molecules, assisted by an anti-freezing salt (CaCl_2)”. What kind of temperature control (T_c or T_f) is the primary purpose of this work?

Response 3-5. Many thanks for the reviewer's good questions. The primary purpose of this work is the temperature control of freezing temperature for water molecules (T_f). Here, T_f could be precisely regulated in a wide range ($0 \sim -58$ °C) by changing the concentration of CaCl_2 (freezing line in Fig. 1a). If not specially stated, the cryogenic temperature (T_c) with -20 °C was used in the manuscript, which is usually adopted in the literatures due to easy obtaining in common refrigerator. However, in order to obtain a detailed data rule and further confirm our proposed mechanism, other cryogenic temperatures ($T_c = -50$ °C, -80 °C) were also tried using a cryogenic medical refrigerator (MDF-U5386S, PHCbi), which could produce the temperature range of $-40 \sim -80$ °C. In order to make it clear, the choice of cryogenic temperature T_c was emphasized in the revised manuscript.

Modification: (Page 12) Here, it should be pointed out that the primary purpose of this work is the freezing temperature control for water molecules via altering CaCl_2 concentration. However, in order to obtain a detailed data rule, other ultralow cryogenic temperatures (T_c

= $-50\text{ }^{\circ}\text{C}$, $-80\text{ }^{\circ}\text{C}$) were also tried using a cryogenic medical refrigerator (MDF-U5386S, PHCbi).

3-6. Fig. S1, cryogel solution with CaCl_2 at $-20\text{ }^{\circ}\text{C}$ seems liquid phase. In this system, salt would be hydrated with water molecules, and would generate ion-dipole interaction between ions and the hydroxyl group of PVA. Fig. 1 is missing the representation of molecular interactions between ions and water.

Response 3-6. Thanks a lot for the reviewer's reminding. We totally agree with the reviewer that the salt would be hydrate with water molecules, transforming some free water molecules into bound water molecules. The schematic illustration of molecular interactions between CaCl_2 and water have complemented in Fig. S1a and the corresponding anti-freezing mechanism was added in the revised manuscript (Page 5).

Additionally, it should be pointed out that the precursor solution with CaCl_2 for suppressed cryogels at $-20\text{ }^{\circ}\text{C}$ for 12 h is transparent, but it represents solid phase. The solid phase could be easily confirmed by its nonflowing state. This appearance is kept after thawing, generating the ultimate hydrogel. In the revised file, in comparison with the flowing state of precursor solution before freezing, its nonflowing state after freezing for arbitrary placement are further supplemented in Supplementary Fig. 1d. In order to see it clear, here the red dye is incorporated into the precursor solution.

Modification: Supplementary Fig. 1d.

Supplementary Fig. 1 d The flowing state of precursor solution and its non-flowing state after freezing at $-20\text{ }^{\circ}\text{C}$ for 12 h.

3-7. Fig. 4, sensory applications experiment, 1 kHz biphasic pulse of 1V amplitude was used to operate the suppressed cryogels-based sensors. How did the authors determine the pulse parameters for the input source? Is there any concern regarding electrolysis or redox reactions?

Response 3-7. Many thanks for the reviewer's questions. For the AC voltage used in the sensory applications experiment in Fig. 4, two kinds of condition (an AC voltage of 1 V

amplitude and 1 KHz frequency (Fig. 4b); an AC voltage of 0.6 V amplitude and 10 KHz frequency (Fig. 4d)) were used to operate the pressure sensor based on the suppressed cryogels. For the selection of pressure sensor input source pulse parameters, as shown in **Supplementary Fig. 17**, we compared the impedance and resistance changes of the pressure sensor under sweep and fixed frequency (the AC amplitude is 0.5 V, 0.6 V, 0.8 V, 1 V with the frequency range of 20 Hz~5 MHz; the AC amplitude is 0.5 V, 0.6 V, 0.8 V, 1 V with the frequency of 10 KHz). It could be seen that with the frequency of 10 KHz, the change of the impedance and resistance of the pressure sensor tends to be stable and the values are the same. Simultaneously, considering the electrolysis phenomenon and based on past experience, an AC voltage with an amplitude of 0.6 V and a frequency of 10 KHz is finally selected to operate the pressure sensor based on the suppressed cryogel. From the results of constant voltage and constant frequency in **Supplementary Fig. 17b-ii** (0.6 V and 10 KHz), it can be seen that no electrolysis phenomenon occurs when the pressure sensor based on the suppressed cryogel is operated with an AC voltage of 0.6 V amplitude and 10 KHz frequency. Besides, for the redox reaction, as shown in **Supplementary Fig. 18**, we tested the CV curves of the suppressed cryogels by cyclic voltammetry, and there was no obvious redox peak, as indicated by the quasi-rectangle in the cyclic voltammetry. Therefore, no redox reaction takes place.

Modification: (Page 18) When operating the pressure sensor based on the suppressed cryogels with an AC voltage of 0.6 V amplitude and a frequency of 10 KHz (Supplementary Fig. 17-18), we found that when the pressure (P) is less than 10 kPa, the sensitivity of the pressure sensor is 101.3 MPa^{-1} , and when $10 \text{ kPa} < P < 100 \text{ kPa}$, the sensitivity is 0.913 MPa^{-1} (Supplementary Fig. 19), with the response time of the pressure sensor being as fast as 18 ms (Supplementary Fig. 20a), which is comparable to other works³⁷⁻⁴⁰.

Supplementary Fig. 17 The impedance and resistance changes of the pressure sensor under sweep and fixed frequency. The impedance and resistance changes of the pressure sensor when the AC amplitude is 0.5 V (a-i), 0.6 V (b-i), 0.8 V (c-i), 1 V (d-i) with the

frequency range of 20 Hz~5 MHz. The impedance and resistance changes of the pressure sensor within 36 s when the AC amplitude is 0.5 V (a-ii), 0.6 V (b-ii), 0.8 V (c-ii), 1 V (d-ii) with the frequency of 10 KHz.

Supplementary Fig. 18 Electrochemical performance of the suppressed cryogel electrodes with different CaCl₂ concentrations. The test was performed at 20 °C with a three-electrode configuration. (a-i, a-ii, b-i, b-ii) Cyclic voltammograms of the suppressed cryogel electrode with a scan voltage of 0~0.7 V and scan rates of 0.05, 0.1, 0.2, 0.5, and 1 V S⁻¹.

3-8. Please double check some typos.

Response 3-8. Many thanks for the reviewer careful review. The grammar and spelling of whole manuscript have been carefully modified and the English language has been polished.

REVIEWERS' COMMENTS

Reviewer #1 (Remarks to the Author):

Although the authors provided a long statement to address my concern, it does not make this work novel. The work is not novel enough for this journal. The authors should submit it to another journal that does not have high requirement on novelty.

Reviewer #2 (Remarks to the Author):

I'm satisfied with the revised manuscript. It can be published as is.

Reviewer #3 (Remarks to the Author):

The authors have fully addressed to our comments. The manuscript can be published without changes.

Response to Referees (NCOMMS-23-03385B)

Reviewer #1:

Comments

Although the authors provided a long statement to address my concern, it does not make this work novel. The work is not novel enough for this journal. The authors should submit it to another journal that does not have high requirement on novelty.

Response: Many thanks for reviewer's careful review. Please allow us to briefly describe the novelty of our work again. Additionally, in order to highlight the novelty of our work, the novelty and our contributions are further emphasized in the revised manuscript.

The brief summary of novelty is shown in the following.

The **green cryogels** have attracted risen attention owing to their remarkable advantages of nontoxicity, simple fabrication without harsh synthetic condition, biocompatibility and their similarity to biological tissues. **However, polymer aggregation-induced semi-crystalline structures make cryogels suffer from extremely restricted functions and sharing the integrated properties (e.g., flexibility, transparency, self-adhesiveness and self-healing) is almost impossible. This situation remains since their discovery in 1970s due to hardly controlled freezing field and the rapid growth of ice crystals.** Therefore, there is an urgent need for the development of the new multi-scale microstructures via a distinctive forming mechanism in cryogels to trigger their structural merits to maximum extent, which would be promising to achieve multifunctionality and satisfy the stringent requirements of emerging applications, such as ionic skin, wearable sensors and optical devices.

In this manuscript, (1) A strategy of **anti-freezing salt assisted suppressed growth of ice crystals** is proposed, which induces **an unusually weak yet homogenous aggregation of polymer chains** at initial freezing gelation stage, due to weakened expelling effect of ice. This distinctive forming mechanism generates **the amorphization of polymer network and simultaneously release part of free supramolecular sites for the first time** due to large distance between adjacent chains. (2) The distinctive multi-scale microstructures endow the simple-recipe cryogels with the **unprecedented integrated properties of tissue-like ultra-softness** (Young's modulus of 4~10 kPa, two magnitude decrease) yet stretchability (~600%), high transparency (~92%), self-adhesion, instantaneous self-healing (<0.3 s) and remoldability, accompanied by superior ionic conductivity, anti-freezing (-58 °C) and water-retention ability. (3) The capacity of using

single suppressed cryogels are demonstrated in a variety of advanced sensory applications, such as **artificial nerve fibers, transparent touch panel and pressure sensor**. Thus our strategy provides both material and conceptual breakthroughs for developing new generation of multifunctional cryogels, which also offers new opportunity in the fields of skin-attachable electronic devices, robotics and prosthetics.

In order to make it clear, the novelty and contributions of our work are further highlighted in the following sections, as marked in red for the revised manuscript.

Paragraph 1 on page 4: “To this end, rather than enhancing polymer aggregation as usual, a distinctive route is proposed: if the ice crystal is suppressed at initial freezing stage of gelation, its expelling effect on polymer chains and the resultant liquid-liquid phase separation (polymer-rich and polymer-poor phases) will be greatly weakened, which is promising to produce a relatively weak yet homogenous aggregation of polymer chains. As a result, the amorphization of polymer network may be achieved and simultaneously release part of free supramolecular sites (e.g., hydroxyl groups) due to large distance between adjacent chains. Based on these particular hierarchical structures at multi-length scales, the potential multifunctionality of cryogels may be triggered based on structure-property theory, which is the interest of our work.”

Paragraph 1 on page 14: “Overall, the emergency of multifunctionality (the ultra-softness, high transparency, self-adhesiveness, instantaneous self-healing, and reprocessing ability) for cryogels with very simple recipe is mainly ascribed to the unique homogenous hierarchical structures, amorphous polymer network, and coexistence of H-bonding/free –OH groups..... Finally, compared with the narrow spacing between adjacent chains anchored by crystalline crosslinked points of cryogels, the amorphous structure endows suppressed cryogels with a large distance between adjacent chains (Fig. 2a), which could serve as rapid transportation channel of ions³⁴⁻³⁶ and remarkably improve ionic conductivity (Supplementary Fig. 12), enlarging their application range as flexible electronic materials (Fig. 4).”

Paragraph 1 on page 20: As a result, an entirely amorphous yet relatively homogenous polymer network is created in freeze-thawed hydrogels, causing a large distance between adjacent chains to make free –OH groups available. In this way, the suppressed cryogels with a simple recipe break the sole semi-crystalline structure and restricted performance of cryogels, representing the unprecedented integrated properties (tissue-like ultra-softness, transparency, self-adhesiveness, instantaneous self-healing, superior ionic conductivity, anti-freezing, etc.).

Reviewer #2:

Comments I'm satisfied with the revised manuscript. It can be published as is.

Response: Many thanks for reviewer's careful review. The useful suggestions helped us a lot to improve the quality of our manuscript.

Reviewer #3:

Comments The authors have fully addressed to our comments. The manuscript can be published without changes.

Response: Many thanks for reviewer's careful review. The useful suggestions helped us a lot to improve the quality of our manuscript.